# EFA6 in Axon Regeneration, as a Microtubule Regulator and as a Guanine Nucleotide Exchange Factor

**DOI:** 10.3390/cells10061325

**Published:** 2021-05-26

**Authors:** Gilberto Gonzalez, Lizhen Chen

**Affiliations:** Barshop Institute for Longevity and Aging Studies, Department of Cell Systems and Anatomy, UT Health San Antonio, San Antonio, TX 78229, USA; gonzalezg11@livemail.uthscsa.edu

**Keywords:** EFA6, axon injury, axon regeneration, microtubule dynamics, integrin, ARF6

## Abstract

Axon regeneration after injury is a conserved biological process that involves a large number of molecular pathways, including rapid calcium influx at injury sites, retrograde injury signaling, epigenetic transition, transcriptional reprogramming, polarized transport, and cytoskeleton reorganization. Despite the numerous efforts devoted to understanding the underlying cellular and molecular mechanisms of axon regeneration, the search continues for effective target molecules for improving axon regeneration. Although there have been significant historical efforts towards characterizing pro-regenerative factors involved in axon regeneration, the pursuit of intrinsic inhibitors is relatively recent. EFA6 (exchange factor for ARF6) has been demonstrated to inhibit axon regeneration in different organisms. EFA6 inhibition could be a promising therapeutic strategy to promote axon regeneration and functional recovery after axon injury. This review summarizes the inhibitory role on axon regeneration through regulating microtubule dynamics and through affecting ARF6 (ADP-ribosylation factor 6) GTPase-mediated integrin transport.

## 1. Introduction

In the adult mammalian central nervous system (CNS), injured axons have a very limited regeneration capacity, resulting in the failure of functional recovery after trauma [1]. This is fundamentally different in the peripheral nervous system (PNS) and in the embryonic CNS, where axons display strong regeneration after injury [2]. Injured mature CNS axons were shown to regrow into sciatic nerve grafts transplanted into the lesion site [3,4], suggesting that CNS axons can regenerate when the environment is permissive, and that the limited CNS axon regeneration is due to the inhibitory environment. It is also becoming evident that CNS axons lose their intrinsic growth ability upon maturation [5], pointing to the importance of understanding the mechanisms underlying the intrinsic axon regeneration abilities of neurons [6,7,8].

The overwhelming evidence, from many organisms, demonstrate numerous conserved genetic and biological mechanisms underlying axon regeneration [1,2,9]. When an axon is severed from an injury, the segment proximal to the soma remains alive and may regenerate in some cases, while the distal segment degenerates [10,11]. Immediately following injury, the plasma membrane at the site of injury is sealed [12,13]. A growth cone is then formed at the proximal axon segment, permitting extension of the axon. This process necessarily involves numerous cellular changes beginning with a rapid influx of calcium at the time of injury from the injury site. Injury signals then travel in a retrograde path from the injury site to the neuronal body, leading to changes in gene expression, cytoskeletal remodeling, and changes in extracellular matrix (ECM) organization [14].

Various injury models have been used to understand the mechanisms of axon regeneration, with spinal cord and optic nerve injury models being the most extensively utilized in the field. Although rodent CNS injury models are employed, non-human primate injury models allow for more clinically relevant investigations [15,16]. Although important, these traditional models model gross CNS injury that necessarily involve amalgamate cellular injuries. This limits the ability to accurately measure axon injury responses at the cell level. A solution to this issue is in utilizing adult PNS neurons as a model for axon injury. These neurons, primarily from dorsal root and sympathetic ganglia, permit investigations of the cellular regenerative program.

In addition to mammalian models, non-mammalian vertebrate (e.g., zebrafish and goldfish) and invertebrate models (e.g., *Drosophila* and *C. elegans*) have long been used. *C. elegans* is a previously validated and powerful model for evaluating the conserved genetic regulation of the nervous system. The extensively documented and simplicity of the *C. elegans* nervous system, combined with its short 3-day lifecycle, make the humble nematode an excellent candidate for molecular and genetic manipulations. The animal itself is transparent, allowing easy visualization and precise laser surgery of single axons labeled with recombinant fluorescent proteins [17]. Ablated axon stumps repeatably regrow, with some variation in response, and will elongate in an error-prone manner [17,18]. This variable and error-prone regrowth may be similar to observed responses in the mammalian CNS subsequent to a laser ablation [19]. The laser axotomy procedure produces a small break in the axon without causing significant damage to the surrounding tissues, permitting study of axon regeneration at the resolution of a single neuron. Unlike mammalian models, the *C. elegans* nervous system is unmyelinated, and the laser microlesions are not associated with persistent scar tissue [17]. Laser assisted axotomies have been utilized in combination with extensive genetic and drug screenings to identify the expression pathways involved in regulating axon regeneration from a traumatic injury [20,21,22]. Within these genetic screens, EFA6 (exchange factor for ARF6) was identified as a negative regulator of axon regrowth [20]. In this minireview, we will provide some background and discuss the function of EFA6 in axon regeneration. Given the inhibitory role of EFA6 in axon regeneration, we suggest that inhibition of EFA6 could be a promising strategy to promote axon regeneration.

## 2. EFA6 Is a GEF for ARF6 and a Microtubule Regulator

ARF6 (ADP-ribosylation factor 6) is a small GTPase that regulates actin remodeling and vesicular transport between the plasma membrane and endosome [23,24]. ARF6 is involved in multiple neuronal processes, including neuron migration, axon and dendrite growth, dendritic spine maturation and maintenance, axonal transport, and synaptic vesicle recycling [25,26,27,28,29,30,31,32]. The spatiotemporal regulation of the GDP/GTP cycle of ARF6 is critical for its diverse functions. The GDP/GTP cycle is controlled by GTPase activating proteins (GAPs) that promote GTP hydrolysis and guanine nucleotide exchange factors (GEFs) that catalyze the exchange of GDP for GTP. The EFA6/PSD (exchange factor for Arf6/pleckstrin and Sec7 domain-containing protein) is one of the GEFs previously identified to activate ARF6 [23]. EFA6 has also been reported to be an efficient GEF for the ARF1 (ADP-ribosylation factor 1) GTPase and its GEF activity for ARF6 and ARF1 are regulated by a negative feedback loop [33].

The mammalian EFA6/PSD family includes four members: EFA6A/PSD1, EFA6B/PSD4, EFA6C/PSD2, and EFA6D/PSD3 [34,35,36,37,38]. In mouse, all four EFA6 proteins are abundantly expressed in the nervous system with overlapping and distinct patterns. Consistent with their expression in the nervous system, defects in EFA6 functions have been linked to neurologic disorders and in human gliomas [39,40]. EFA6 proteins are characterized by a conserved Sec7 domain (GEF activity domain), a pleckstrin homology (PH) domain that permits membrane localization as well as interaction with actin filaments, a C-terminal coiled-coil domain for protein–protein interaction, and an N-terminal intrinsically disordered domain (Figure 1) [36,41]. In *Drosophila* and in *C. elegans,* there is only one Efa6 gene.

The *C. elegans* efa-6 gene was initially identified as a suppressor gene from a screen for the lethality associated with conditional dynein heavy chain (*dhc-1*) mutations [42]. In one-cell stage *C. elegans* embryos, spindle positioning and function depends on the contact of microtubules with the cell cortex [43]. The microtubule minus-end-directed motor dynein regulates spindle position and loss of dynein leads to embryonic lethality [44,45]. Loss of EFA-6, a cortically localized protein, leads to abnormally long and dense microtubules at the cell cortex in early embryos. In *efa-6* mutants, growing microtubule plus ends, visualized by a labeled plus end binding protein, resided at the cortex for longer than wildtype. Surprisingly, they found that the function of EFA-6 in regulating microtubules was dependent only on its N-terminus, independent of ARF-6 activity. Within this intrinsically disordered N-terminus, a conserved 18 amino acid (a.a.) motif was identified in both nematodes and arthropods (Figure 1), while the rest of the N-terminus showed little sequence homology in different species.

A recent study on *Drosophila* Efa6 has revealed that it is widely expressed in the *Drosophila* nervous system and can restrict axonal growth [46]. Efa6 functions as a cortical collapse factor acting through the conserved 18 a.a. motif at the N-terminus, which they named N-terminal microtubule elimination domain (MTED). They showed that the MTED bound tubulin and inhibited microtubule polymerization in vitro through a direct interaction between Efa6 and tubulin, while the C-terminal domain restricted its microtubule inhibiting activity to the cortex.

## 3. Microtubule Dynamics in Axon Regeneration

### 3.1. Microtubules in Intact Axons

Microtubules are polarized cylindrical polymers, composed of α- and β-tubulin heterodimers that form protofilaments [47]. They undergo periods of polymerization and depolymerization primarily at their plus ends. The minus ends are relatively stable compared to plus ends. A larger number of microtubule binding proteins have been identified to associate with the microtubule plus end. Relatively fewer minus end binding proteins have been found. Microtubule dynamics are briefly defined as the amalgamate processes involved in microtubule growth and shrinkage. Switching from growth to shrinkage is defined as a microtubule catastrophe and switching from shrinkage to growth is defined as a rescue.

Microtubule dynamics are also affected by the posttranslational modifications (PTM) of tubulins, including acetylation, detyrosination, and Δ2 modification, after their incorporation into the microtubule polymer [48]. Older microtubules often collect more modifications, which might conversely contribute to the stability of these microtubules. Acetylation of α-tubulins is the only luminal microtubule PTM and is associated with stable microtubules. The conserved EEY motif at the C-terminus of α-tubulin is a substrate for detyrosination and deglutamylation. The subterminal glutamate of detyrosinated tubulin can be removed by cytosolic carboxypeptidase (CCPs). The resulting Δ2-tubulin is resistant to microtubule depolymerizing motors, resulting in microtubule stabilization [49].

Cellular compartments undergoing morphological changes, such as growing axons, are associated with dynamic microtubules [50]. Microtubules within mature neurons are significantly more stable than during neuronal development. The microtubules within the axonal shaft in particular are substantially more stabilized than in other parts of the neuron [14]. In addition, they demonstrate polarity within the neuron. Plus ends are pointed toward the distal axon from the soma and dendritic microtubules are either minus or plus orientated from the soma [51]. The microtubule orientation is critical for polarized trafficking to axons and dendrites by supporting the directed movement of microtubule motor proteins [52]. Microtubules undergo dynamic changes, and their stability can be regulated by a large number of microtubule regulators.

The extensive role of cytoskeletal dynamics in axon regeneration is without question. In particular, the role of both microtubules and actin filaments are crucial in axon and growth cone formation [53,54]. This relationship begins with the initial differentiation of neurite branches into axons that is in part triggered by actin depolymerization and microtubule stabilization [55,56,57,58]. The evidence for this exists with treating developing neurons with Taxol (a known microtubule stabilizing drug) and cytocholasin (a known actin destabilizing drug) inducing multiple axon formation within the neuron [59].

Both developing and regenerating axons consist of an elongating shaft as well as a dynamic growth cone [53,54,60]. The axon shaft is composed primarily of stabilized microtubules [59,61]. The growth cone is composed of two domains, a peripheral actin rich domain and a central domain with dynamic microtubules [54,57]. The peripheral domain has radial and circumferential actin filaments forming the filopodium and lamellipodium, respectively. The central domain contains the plus ends of actively growing microtubules. The filopodial actin filaments first extend to allow growth cone movement [57,58,62]. Subsequently, the proximal ends of the filopodial actin filaments will be destabilized, allowing a few microtubules to breach the central domain into the peripheral actin rich domain alongside filopodial actin [56]. These explorative microtubules are then stabilized, and the axon is extended forward [63,64]. This process repeats until the growing axon reaches its destination. This process requires rigid orchestration, with deviations from ideal dynamics resulting in significantly altered axon growth capacity [63,65,66].

### 3.2. Microtubules in Injured and Regenerating Axons

Following neuronal injury, previously stabilized microtubule bundles along the axon are broken down and reorganized to facilitate bidirectional intracellular transport [67]. This reorganization facilitates the retrograde movement of injury signals, altering expression patterns to promote regrowth, as well as the anterograde movement of materials for the elongating axon. Reorganization of axonal microtubules post injury was first demonstrated in cultured *Aplysia* neurons [68,69]. Using an EBP-GFP (microtubule plus end binding protein fused with GFP) reporter and live imaging, they showed that axon injury disrupted microtubule arrays locally. Microtubules at the proximal axon stump were then rapidly depolymerized, followed by repolymerization and generation of a microtubule-based “trap” with microtubules of mixed polarity at ~100 μm regions from the injury sites. They also showed that plus-end directed vesicles that contain materials for regenerative growth cone formation accumulated in the trap [69]. Notably, microtubule destabilization drugs were able to block the formation of regenerative growth cone after injury [70], supporting that microtubule reorganization post axonal injury is critical for regeneration initiation.

Axon injury-triggered rearrangements in the microtubule cytoskeleton are also found in *Drosophila* sensory neurons [71,72]. Axotomy at 20~50 μm away from the cell body triggers reversal of microtubule polarity and leads to regrowth from the axon stump. Interestingly, axon injury adjacent to the soma results in global up-regulation of growing microtubules in the soma and polarity reversal of dendritic microtubules, which converts the dendrite to a regrowing axon-like process.

In *C. elegans* touch sensory neurons, axotomy also induces microtubule re-organization. Axotomy triggers a rapid down regulation of growing microtubules in both proximal and distal fragments within seconds post injury [73,74]. The number of growing microtubules increases locally at the proximal stump in 2–3 h post injury, although the overall catastrophe frequency, growth duration, and velocity remain similar to those of intact axons [73]. At 3–6 h post axotomy microtubule catastrophe frequency is decreased without affecting growth velocity, resulting in prolonged microtubule growth [73], and this is associated with the initiation phase of axon regeneration [20]. At 10–14 h, increased microtubule growth velocity is observed, coincident with the fast axon extension rate at this phase. Differentiating *C. elegans* from *Aplysia* or *Drosophila*, reversal of microtubule polarity is only rarely observed proximate to the injury site after axotomy [73].

Axon growth following injury proceeds similarly to its initial formation. However, within the mammalian central nervous system (CNS), this does not occur [66,75,76,77]. The amalgamate evidence suggests that dysregulation of cytoskeletal dynamics is a major cause of the inability of the mammalian CNS to regenerate following injury [76,77,78]. This is clearly noted by the formation of retraction bulbs following a traumatic axon injury [79]. Retraction bulbs, round structures forming at the proximal axonal stump following injury, demonstrate disorganized networks of microtubules [76,77,78]. Treatment with microtubule stabilization agents (such as Taxol) prevent retraction bulb formation [79,80,81]. Additionally, treatment of growing axons with microtubule stabilizers improves axon regeneration [80]. If nocodazole (a microtubule destabilizing agent) is given instead, it induces a conversion of the axon tip into a retraction bulb [82], again demonstrating the importance of microtubule orchestration for axon regenerative capacity.

## 4. ARF6, Integrins, and Axon Regeneration

Sensory axons that extend from the peripheral nervous system (PNS) into the spinal cord have regeneration capacity after injury. However, their regrowth is often limited by inhibitory molecules in the spinal cord, such as tenascin-C, an extracellular matrix (ECM) molecule [83]. Integrins are transmembrane receptors that recognize molecules within the ECM [84]. Viral expression of an integrin that recognizes tenascin-C, α9, β1, along with its activator kindlin-1 in sensory neurons has been shown to promote long-range regeneration of sensory axons and functional recovery by interacting with ECM components that play inhibitory roles in axon regeneration [85].

Integrins transduce extracellular cues to the cytoskeleton, activating internal signaling pathways that can regulate various biological processes including cell survival, morphology, motility, and the cell cycle. The integrin family consists of 18 α and 8 β subunits, which form 24 hetero-dimeric cell surface receptors. Integrins are widely expressed in the central nervous system (CNS) and their functions depend on their cellular localization. During embryonic development, integrins are localized to axons in most neurons. In the adult nervous system, integrins are excluded from the axons of many CNS neurons but are maintained in axons of retinal and sensory neurons, two neuronal subtypes that demonstrate higher regenerative capacity relative to many other CNS neuronal subtypes [10,86,87,88,89], indicating the correlation between axonal localization of integrins and axon regeneration.

After a PNS nerve injury, the composition of the ECM undergoes changes, with collagen, fibronectin, and laminin becoming the major components of the basal lamina [90]. These ECM molecules generate a microenvironment that supports cell adhesion and axon regeneration. Up-regulation of integrins have been observed after nerve injury in the peripheral nervous system (PNS) and integrins are considered regeneration-associated genes [91,92,93,94,95,96]. The importance of integrins in axon regeneration is supported by knockout studies. Knockout of several integrin subunits diminished peripheral nerve regeneration, although it remains unclear whether a single knockout will inhibit regeneration in the PNS, as many integrins are present in the axons and they recognize several ligands. Mice deficient in integrin α7 showed reduced facial and sciatic nerve regeneration after axotomy [92,95]. In addition, inhibiting α7 and β1 function using function-blocking antibodies resulted in impaired neurite outgrowth of cultured dorsal root ganglia (DRG) following a conditioning lesion in vivo [92,97].

The localization of integrins in PNS axons appears critical for its role in promoting axon regeneration, and exclusion of integrins from adult CNS axons might contribute to the failure of CNS axon regeneration. Therefore, understanding the mechanisms of integrin trafficking and transport will provide insights for promoting CNS axon regeneration. Selective polarized transport mechanisms in CNS neurons ensure axon and dendrite-specific localization of molecules essential for proper axon and dendrite function [98,99,100]. As a result of the polarized transport, integrins are excluded from mature CNS axons [101,102]. The axonal distribution of integrins during maturation has attracted intensive attention, as any integrin-based attempt to promote axon regeneration requires targeting integrins to the axon and growth cone. In cancer cells, integrins are transported in recycling endosomes under the control of small GTPase [103]. In neurons, axonal integrins are often transported in recycling endosomes that are associated with small GTPases Rab11 and ARF6 [26,103,104,105]. Two known guanine exchange factors (GEFs) for ARF6 are ARNO (ARF nucleotide-binding site opener) and EFA6, which have been reported to be up-regulated during cortical neuron maturation [34,101]. Both ARNO and EFA6 play important roles in the exclusion of integrins from axons. The direction of transport is determined by the activation state of the small GTPase. Vesicles bound to GTP forms of ARF6 and Rab11 are retrograde transported, while vesicles bound to ARF6- and Rab11-GDP move in the anterograde direction [106].

## 5. EFA6 as an Inhibitor of Axon Regeneration

### 5.1. EFA6 Inhibits Axon Regeneration through Regulating Microtubule Dynamics

Our previous genetic screen for pathways regulating axon regeneration in *C. elegans* identified EFA-6 as an intrinsic inhibitor for axon regeneration [20]. Focusing on genes that are not essential for overall health or growth, we examined genetic mutants for genes that are conserved between *C. elegans* and human and have been implicated to function in the nervous system. To examine axon regeneration, we perform laser axotomy using the touch sensory PLM (posterior lateral microtubule) neurons, which display consistent and robust axon regrowth after injury [18]. We measured the length of regenerated axons 24 h post laser injury and compared the regrowth length in mutants to a wildtype control. We found that genes affecting PLM regrowth are among all structural and functional gene clusters tested, including cell adhesion/ECM, gene expression, neurotransmission, cytoskeleton, trafficking, protein homeostasis, and signaling transduction [20].

*efa-6* was among the few genes with inhibitory effects on regrowth identified from the genetic screen. *efa-6* mutants are viable and display mild PLM axon overshooting in development. Axon regeneration length in *efa-6* mutants was also significantly longer than in wildtype. Cell type-specific transgenic expression of EFA-6 in all neurons or in touch neurons, but not in muscles, rescued the developmental axon overshooting defect and inhibited axon regeneration after injury, indicating that EFA-6 functions cell-autonomously. To understand the mechanism by which EFA-6 inhibits axon regrowth, we first examined the *arf-6* mutants, as EFA-6 is a known GEF for ARF-6 [33]. Similar to *efa-6* mutants, *arf-6* mutants also displayed enhanced axon regrowth, suggesting an inhibitory role of ARF-6 in axon regeneration. We also found that EFA-6 overexpression potently inhibited axon regeneration in *arf-6* mutants, indicating that the inhibitory effect of EFA-6 is not fully mediated by ARF-6. Transgenic expression of mutant EFA-6 proteins lacking the Sec7 domain or with mutated catalytic residue within the Sec7 domain was sufficient to inhibit axon regrowth, further implicating a GEF-independent role for EFA-6 in inhibiting axon regeneration. Further investigations revealed that mutant EFA-6 lacking the N-terminus failed to inhibit regrowth. Additionally, the N-terminus alone was sufficient to inhibit axon regrowth in a dose-dependent manner.

As mentioned previously, EFA-6 is known to regulate microtubule growth in *C. elegans* embryos through its N-terminal domain [42]. Epistasis analysis suggested that *efa-6* might function in the same pathway with *ebp-1* [20], which encodes a microtubule plus end binding protein. Notably, the effects of EFA-6 overexpression could be partially overcome by injection of Taxol [20], suggesting that EFA-6 might act as an endogenous catastrophe factor in the axonal MT cytoskeleton. To examine whether EFA-6 regulates axonal microtubules, we examined microtubule dynamics marked by the plus end binding protein EBP-2 [20,74]. In intact PLM axons of *efa-6* mutants, there were more growing microtubules compared to control, while in EFA-6 overexpressing neurons the number of growing microtubules was significantly reduced. Immediately post injury, axonal EBP-2::GFP comets were dramatically reduced in wildtype animals. This injury-induced reduction of growing microtubules might be due to injury-triggered Ca^2+^ transients, as Ca^2+^ is known to promote microtubules disassembly [107,108,109,110,111]. Ca^2+^ can directly bind to microtubule and weaken tubulin–tubulin interaction [112]. Injury-induced Ca^2+^ transients have also been shown to traverse the axon shaft to a growth cone in cultured *Aplysia* neurons [113]. A change in acute injury-induced EBP-2::GFP comets was not significant in *efa-6* mutants. At 3 h post axotomy, growing microtubules were increased, and this was further enhanced in *efa-6* mutants, coincident with the enhanced axon regrowth at the initiation phase in *efa-6* mutants [74]. Taken together, the data indicates that EFA-6 directly functions of EFA-6 in regulating axonal microtubules in response to injury.

EFA-6 was further found to interact with the microtubule associated proteins TAC-1, the *C. elegans* ortholog of human TACC (transforming acidic coiled-coil), and ZYG-8, the *C. elegans* ortholog of human DCLK (double-cortin-like kinase), through its N terminus [74]. Although TAC-1 and ZYG-8 are centrosomal proteins, they can both localize in axons and are each required for axon regeneration [74]. *Xenopus* TACC3 was shown to promote axon outgrowth in embryonic cultured neural crest cells [114], and DCLK is required for axon regrowth in mice [115]. Compared to wildtype, *tac-1* and *zyg-8* mutants demonstrated fewer growing microtubules in uninjured axons, and do not exhibit reduced microtubule dynamics in response to injury. Epistasis analyses suggested that both TAC-1 and ZYG-8 regulate microtubule dynamics and promote axon regeneration downstream to EFA-6.

EFA-6 is normally localized to the cell membrane in intact mature neurons through its PH (pleckstrin homology) domain [74]. Within seconds post axon injury, EFA-6 displays a rapid trans-localization to punctate structures. The rapid translocation of EFA-6 and TAC-1 correlates closely with the altered microtubule dynamics triggered by axon injury. EFA-6 translocation is independent of membrane association, as EFA-6 lacking the PH domain as well as the N-terminus alone, which themselves are not membrane-localized, re-localizes to axonal puncta after injury. In contrast, the 18 a.a. motif within the N-terminus was found essential for the injury-induced trans-localization [74]. TAC-1 also displayed similar trans-localization in response to injury. Interestingly, EFA-6 and TAC-1 puncta both co-localize with puncta that are labeled with the microtubule minus end protein PTRN-1/CAMSAP, suggesting that microtubule minus ends might be a site of regulation by EFA-6 in axon regeneration after injury. Thus, these studies uncover a mechanism by which EFA-6 influences microtubule dynamics to regulate axon regeneration (Figure 2).

### 5.2. EFA6 Inhibits Axon Regeneration as a GEF for ARF6

As discussed earlier, integrins are important for axon regeneration. However, they are excluded from the mature central nervous system (CNS) due to both reduced anterograde and enhanced retrograde integrin transport in the axon [106]. In cultured cortical neurons derived from E18 (embryonic day 18) rat embryos, the expression levels of α5, αV and β1 integrins were found to start decreasing at DIV7 (day in vitro 7) and were completely lost in the axon at DIV14 [101]. The exclusion of integrins from the axon is associated with the formation of the axon initial segment (AIS) [116]. The AIS is a specialized compartment in neurons that is positioned in the proximal axon. It divides the axonal and somatodendritic domains. The AIS serves two important functions within neurons: action potential firing and neuron polarity maintenance [117]. A dense network of cytoskeletal proteins is localized within the AIS. The cytoskeletal proteins within the AIS can limit access of molecules to axons by serving as both a size filter and supporting retrograde transport [116,118,119]. The AIS is believed to play an important role in the axonal exclusion of integrins, as its disruption leads to an increased integrin level in mature CNS axons [101].

EFA6A has been demonstrated to be enriched in the initial part of the axon, co-localized with neurofascin, a marker of AIS, in DIV7 rat cortical neurons [120]. Unlike in invertebrate neurons, silencing EFA6A with shRNA did not affect axonal microtubule dynamics as measured by EB3-GFP (microtubule end binding protein 3 fused with GFP) comets. This is consistent with the lack of an MTED in mammalian EFA6 proteins. Using a reporter for ARF activity, ABD-GGA3, the study also showed that active ARF protein is distributed throughout axons at DIV4 and DIV14, two time points before and after EFA-6A is enriched in AIS. Removing EFA6A with shRNA strongly reduced active ARF6 in the axon without affecting the total amount of ARF6. Therefore, despite being enriched in AIS, EFA6A can activate ARF6 throughout axons. Given that the direction of axonal integrin transport is regulated by the activation state of ARF6, with ARF6-GTP promoting retrograde transport and ARF6-GDP facilitating anterograde transport, depletion of EFA6A, the GEF for ARF6, could be effective to enhance anterograde integrin transport and to promote axon regeneration. Indeed, depleting EFA6A with shRNA resulted in initiation of anterograde transport, reduced retrograde transport and enhanced integrins throughout the axon. This is in contrast to the control shRNA treated neurons, in which anterograde transport was barely detectable [120]. Using an in vitro axotomy system, the study also reported that cultured embryonic cortical neurons transfected with EFA6 shRNA displayed substantial increase in regeneration compared to neurons transfected with control shRNA.

Consistent with the inhibitory role of EFA6 in axon regeneration, EFA6 level in adult dorsal root ganglion (DRG) neurons is high in the cell body but low throughout the axon [120]. Cultured adult DRG neurons display robust axon regeneration post laser axotomy. However, when EFA6 is overexpressed in DRG neurons, the regeneration capacity, measured by the presence of regenerative growth cones, is dramatically reduced. Although, DRG neurons overexpressing EFA6E242K, a mutant with abolished Sec7 GEF activity, show much better axon regeneration compared to wildtype EFA6-overexpressing neurons, suggesting that the inhibitory function of EFA6 is mediated by its GEF activity towards ARF6. Together, these studies demonstrated a pathway through which EFA6 controls ARF6 activity to regulate integrin transport and axon regeneration (Figure 3).

## 6. Conclusions

Despite recent advance in identification of axon regeneration regulators, the mechanisms underlying axon regeneration are not yet completely understood. Historically, much effort has been devoted to identifying proregenerative factors. But neuron intrinsic factors that inhibit axon regeneration have attracted less attention for decades until recently. EFA6 is a conserved protein that is known to act as a GEF for ARF6 GTPase. EFA6 was first identified from a genetic screen in *C. elegans* as an intrinsic inhibitor for axon regrowth. However, its role in inhibiting *C. elegans* axon regrowth was independent of its GEF activity but required an 18 a.a. motif within its otherwise disordered N-terminus. Further studies in *C. elegans* and *Drosophila* neurons revealed the role of EFA6 in regulating axonal microtubule dynamics for axon development and regeneration. In mammalian neurons, EFA6 was additionally found to inhibit axon regeneration through activating ARF6, which plays an important role in integrin transport. Although the N-terminal 18 a.a. motif that is important for microtubule regulation is conserved between nematodes and insects, it is absent from mammalian EFA6 proteins. It is not clear how this motif was lost during evolution. As EFA6 can inhibit axon regrowth both by influencing axonal microtubule dynamics and by regulating ARF6-mediated transport, inhibition of EFA6 could be a promising strategy to promote axon regeneration and functional recovery after injury or in neuronal diseases.

## Figures and Tables

**Figure 1 cells-10-01325-f001:** The conserved EFA6 protein family. (**A**) Graphic illustration of EFA6 family proteins showing the PH (Pleckstrin Homology), Sec7 (GEF activity) and CC (coiled coil) domains, based on sequences from NCBI. Not to scale. An 18 a.a. motif (microtubule elimination domain, MTED) is present in *C. elegans* and *Drosophila* EFA6 proteins, but absent in mammalian EFA6 proteins. There are four EFA6 paralogs in mammals (EFA6A-D) and the length of their N-termini varies. (**B**) Plot of intrinsic protein disorder probability for *C. elegans* EFA-6 using PrDos (http://prdos.hgc.jp/cgi-bin/top.cgi), access date 23 May 2021. The N-terminus is overall highly disordered except for the 18 a.a. motif. The domains are highlighted using the same color code as (**A**).

**Figure 2 cells-10-01325-f002:** EFA-6 regulates axonal microtubule dynamics to inhibit axon regeneration. Axon injury of *C. elegans* neurons triggers a rapid relocalization of EFA-6 from the plasma membrane to structures overlapping with the microtubule minus end binding protein PTRN-1. This relocalization of EFA-6 occurs simultaneously with an injury-induced local downregulation of axonal microtubule polymerization, a process that requires EFA-6. EFA-6 directly interacts with TAC-1 and ZYG-8, two conserved microtubule regulators that are required for axon regeneration. At 2–3 h post injury, EFA-6 protein returns back to the plasma membrane, coincident with an up-regulation of microtubule polymerization.

**Figure 3 cells-10-01325-f003:** EFA6 inhibits axon regeneration by activating ARF6 to regulate integrin transport. EFA6 is enriched in AIS and can activate ARF6 throughout the axon. ARF6-GTP facilitate retrograde transport of integrin, resulting in exclusion of integrin from the axon and limited axon regeneration. EFA6 knockdown leads to reduced ARF6 activity. ARF6-GDP promotes anterograde transport of integrin, enhancing axonal integrin level and axon regeneration capacity.

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
