# Peer review of "RETRACTED: EFA6 in Axon Regeneration, as a Microtubule Regulator and as a Guanine Nucleotide Exchange Factor"

_cells, 2021, doi:10.3390/cells10061325_

Round 1
Reviewer 1 Report
have got a chance to look over the review. Generally it is pretty good and should move forward. I have a few suggestions that will hopely help clarify the manuscript.
- Figure IA and manuscript: There was a indication for amino acid as "aa" with different spacing in everywhere in Figures and MS. It should be modified as amino acid or a.a. with same spacing.
- Figure IB : This figure is exactly same in the author's previous publication in eLife(2015). I am not sure whether it is O.K. or not. Please check this issue.
- There is an overusage of acronyms which makes it hard to understand the point often times. I think that most of readers may not be very familiar with the level of detail that the authors are given the novel subject matter. I would suggest that the authors should provide a full name of terms when they appears first time in the each sections. For example, GTPase, ARF6, PSD, EBP-GFP,PLM, DRG etc...
Author Response
Response to Review#1:
have got a chance to look over the review. Generally it is pretty good and should move forward. I have a few suggestions that will hopely help clarify the manuscript.
--We thank the reviewer for the positive comments and for the valuable suggestions!
1. Figure IA and manuscript: There was a indication for amino acid as "aa" with different spacing in everywhere in Figures and MS. It should be modified as amino acid or a.a. with same spacing.
--As suggested, we have replaced all “aa” with “a.a.”.
2. Figure IB : This figure is exactly same in the author's previous publication in eLife(2015). I am not sure whether it is O.K. or not. Please check this issue.
--We appreciate the comment and have generated a new graph for Figure 1B.
3. There is an overusage of acronyms which makes it hard to understand the point often times. I think that most of readers may not be very familiar with the level of detail that the authors are given the novel subject matter. I would suggest that the authors should provide a full name of terms when they appears first time in the each sections. For example, GTPase, ARF6, PSD, EBP-GFP,PLM, DRG etc...
--We appreciate the comment and have provided a full name for each term when it appears the first time in each section.

Reviewer 2 Report
The authors discussed the role of the exchange factor for ARF6 (EFA6) in axon regeneration with a particular focus on the regulation of microtubule dynamics and integrin transport. The review is easy to read and well organized.
Minor comments:
- The authors may consider rephrasing the following sentence in the abstract “suggesting that EFA6 could be a promising therapeutic target for promoting axon regeneration and functional recovery after axon injury”. In its current form, the general reader may understand EFA6 acts as a pro regenerative factor. To avoid any misunderstanding, the authors may consider the following: “EFA6 inhibition may be a promising therapeutic strategy to promote axon regeneration and functional recovery after axonal injury”.
- Historically, much more effort has been devoted to identify proregenerative factors. Neuron intrinsic factors that inhibit axon regeneration have been largely unnoticed (or simply attracted less attention) for decades until recently. The authors may want to stress this concept when introducing EFA6 as an inhibitor of axon regeneration.
- When introducing the laser axotomy, it should be clear that such experimental model it not causing scarring.
- Figure 1: the authors may consider including representative structures from mice and rats as well.
- In section 3.1 “Microtubules in intact neurons” The authors may consider adding some wording on microtubules post translational modifications (in particular acetylation, tyrosination and detyrosination).
- A few spelling errors: line 98 “EFA-6 in regulating microtubules was was dependent only its N-terminus”; line 336 “at DIV7 and were completed lost in the axon at DIV14”
Author Response
Response to comments from Reivew#2
Review 2 Comments
The authors discussed the role of the exchange factor for ARF6 (EFA6) in axon regeneration with a particular focus on the regulation of microtubule dynamics and integrin transport. The review is easy to read and well organized.
--We thank the reviewer for the positive comments!
Minor comments:
1. The authors may consider rephrasing the following sentence in the abstract “suggesting that EFA6 could be a promising therapeutic target for promoting axon regeneration and functional recovery after axon injury”. In its current form, the general reader may understand EFA6 acts as a pro regenerative factor. To avoid any misunderstanding, the authors may consider the following: “EFA6 inhibition may be a promising therapeutic strategy to promote axon regeneration and functional recovery after axonal injury”.
-- We appreciate the comment and have made changes accordingly.
Lines16-17: EFA6 inhibition could be a promising therapeutic strategy to promote axon regeneration and functional recovery after axon injury.
2. Historically, much more effort has been devoted to identify proregenerative factors. Neuron intrinsic factors that inhibit axon regeneration have been largely unnoticed (or simply attracted less attention) for decades until recently. The authors may want to stress this concept when introducing EFA6 as an inhibitor of axon regeneration.
-- We greatly appreciate the point raised by the reviewer and have emphasized the role of EFA6 as an intrinsic inhibitor.
Lines 13-16: Although there have been significant historical efforts towards characterizing pro-regenerative factors involved in axon regeneration, the pursuit of intrinsic inhibitors is relatively recent.
Lines 430-433: Historically, much effort has been devoted to identifying proregenerative factors. But neuron intrinsic factors that inhibit axon regeneration have attracted less attention for decades until recently.
3. When introducing the laser axotomy, it should be clear that such experimental model it not causing scarring.
-- As suggested, we have included the discussion that such laser axotomy model does not cause scarring.
Lines 62-66: Laser axotomy produces a small break in the axon without causing significant damage to the surrounding tissues, permitting study of axon regeneration at the resolution of a single neuron. Unlike the mammalian models, the C. elegans nervous system is unmyelinated, and laser microlesions are not associated with persistent scar tissue [17].
4. Figure 1: the authors may consider including representative structures from mice and rats as well.
-- As suggested, we have added structures of mouse and rat EFA6 proteins in Figure 1A.
5. In section 3.1 “Microtubules in intact neurons” The authors may consider adding some wording on microtubules post translational modifications (in particular acetylation, tyrosination and detyrosination).
-- We appreciate the reviewer for the suggestion. We have now included microtubule post translational modification in section 3.1.
Lines 142-157: Microtubule dynamics is also affected by the posttranslational modifications (PTM) of tubulins, including acetylation, detyrosination and Δ2 modification, after their incorporation into the microtubule polymer [48]. Older microtubules often collect more modifications, which might conversely contribute to the stability of these microtubules. Acetylation of α-tubulins is the only luminal microtubule PTM and is associated with stable microtubules. The conserved EEY motif at the C-terminus of α-tubulin is a substrate for detyrosination and deglutamylation. The subterminal glutamate of detyrosinated tubulin can be removed by by cytosolic carboxypeptidase (CCPs). The resulting Δ2-tubulin is resistant to microtubule depolymerizing motors, resulting in microtubule stabilization [49].
6. A few spelling errors: line 98 “EFA-6 in regulating microtubules was was dependent only its N-terminus”; line 336 “at DIV7 and were completed lost in the axon at DIV14”
-- We apologize for the spelling errors and have corrected the errors.
